# Anti-Ku Antibodies: Clinical Associations, Organ Damage, and Prognostic Implications in Connective Tissue Diseases

**DOI:** 10.3390/ijms26157433

**Published:** 2025-08-01

**Authors:** Céline La, Julie Smet, Carole Nagant, Muhammad Soyfoo

**Affiliations:** 1Department of Rheumatology, Hôpital Erasme, Hôpital Universitaire de Bruxelles (HUB), 1070 Bruxelles, Belgium; celine.la@ulb.be; 2Laboratory of Immunology, Laboratoire Hospitalier Universitaire de Bruxelles, Université Libre de Bruxelles (LHUB-ULB), 1020 Bruxelles, Belgium; julie.smet@chu-brugmann.be (J.S.); carole.nagant@chu-brugmann.be (C.N.)

**Keywords:** anti-Ku antibodies, connective tissue diseases, interstitial lung disease, autoantibodies, prognosis, borderline results

## Abstract

Anti-Ku antibodies are rare autoantibodies associated with connective tissue diseases (CTDs), but their clinical significance remains poorly understood due to limited studies. Semi-quantitative immunodot assays yield positive, negative, or borderline results, with the clinical relevance of borderline findings remaining unclear. The purpose of this study is to characterize the clinical spectrum of anti-Ku-positive patients and evaluate the clinical significance of anti-Ku-borderline results in CTD management. A retrospective cohort study was conducted at Hôpital Erasme, including all patients with anti-Ku-positive or borderline results, over a 10-year period. Clinical and biological data were collected from medical records and analyzed for disease associations, organ involvement, and outcomes. Among 47 anti-Ku-positive patients, systemic lupus erythematosus (SLE) and Sjögren’s syndrome (SS) were the most common diagnoses. Interstitial lung disease (ILD) occurred in 23.4% and renal involvement in 12.8% of patients. Cytopenia was significantly associated with glomerulonephritis. Organ damage, particularly pulmonary and renal involvement, correlated with increased mortality. In the borderline group (n = 33), SLE and SS remained the predominant diagnoses. During follow-up, three patients died (all with isolated ILD without associated CTD), one required chronic dialysis, and one underwent lung transplantation. ILD was present in 7/22 (31.8%) borderline patients, and renal involvement in 7/32 (21.9%). This study demonstrates significant associations between anti-Ku antibodies and organ damage, with increased mortality risk. The high prevalence of pulmonary and renal involvement in anti-Ku-borderline patients suggests that these results carry substantial clinical significance and should prompt comprehensive CTD evaluation. These findings support treating borderline anti-Ku results with the same clinical vigilance as positive results, given their similar association with severe organ involvement and adverse outcomes.

## 1. Introduction

The Ku protein is a DNA-binding heterodimer composed of Ku70 and Ku80, playing a central role in DNA double-strand break repair via the classical non-homologous end-joining (c-NHEJ) pathway. By rapidly binding DNA ends, Ku protects genomic integrity and facilitates the recruitment of essential repair factors. Beyond DNA repair, Ku also contributes to telomere stability, V(D)J recombination, and cellular homeostasis [1,2,3].

Anti-Ku antibodies were first identified in 1981 in a patient presenting with inflammatory myositis, Raynaud’s phenomenon, and skin thickening of the upper extremities [1]. This discovery marked the beginning of our understanding of a unique autoantibody system that would prove to have significant clinical implications across multiple connective tissue diseases. Subsequent biochemical investigations revealed that these autoantibodies target a heterodimeric antigen complex composed of 70 kDa and 80 kDa proteins (Ku70/Ku80), which forms the DNA-binding component of the DNA-dependent protein kinase (DNA-PK) complex [2,3,4]. This complex plays a crucial role in cellular DNA repair mechanisms, particularly in the non-homologous end-joining pathway for double-strand break repair, and is essential for maintaining genomic stability and cellular survival [5,6].

The Ku antigen complex demonstrates remarkable evolutionary conservation across species, highlighting its fundamental importance in cellular biology. The heterodimer binds with high affinity to double-stranded DNA breaks, serving as a scaffolding protein that recruits additional DNA repair machinery to sites of genomic damage [7]. This molecular function has led to the speculation that autoimmune targeting of the Ku complex may result from molecular mimicry or the exposure of cryptic epitopes during cellular stress or apoptosis, potentially explaining the association with inflammatory conditions [8,9].

Initially, anti-Ku antibodies were considered pathognomonic for the systemic sclerosis (SSc)–polymyositis (PM) overlap syndrome [1]. This initial association led to their classification as a marker of overlap syndromes, a concept that has since evolved considerably. However, subsequent clinical studies have demonstrated their presence across numerous connective tissue diseases (CTDs), fundamentally changing our understanding of their diagnostic significance. These include systemic lupus erythematosus (SLE), Sjögren’s syndrome (SS), idiopathic inflammatory myopathies (IIMs), SSc, rheumatoid arthritis (RA), psoriatic arthritis (PsA), and various overlap syndromes [10,11,12,13,14,15,16,17,18]. The breadth of associated conditions suggests that anti-Ku antibodies may represent a common pathogenic pathway or response pattern rather than a disease-specific marker.

The clinical presentation of anti-Ku-positive patients varies significantly depending on the underlying CTD. In SLE, these antibodies have been associated with more severe disease manifestations, including nephritis and neurological involvement [19,20]. In myositis, anti-Ku antibodies often correlate with milder muscle inflammation but increased risk of pulmonary complications [21]. The heterogeneity of clinical presentations has made it challenging to establish unified diagnostic criteria or treatment protocols for anti-Ku-positive patients across different CTDs.

Notably, the prevalence and clinical associations of anti-Ku antibodies demonstrate significant ethnic and geographic variation. Wang et al. demonstrated higher frequency in African Americans with SLE compared to Caucasian patients [22], while other studies have reported varying prevalences across different populations, suggesting that genetic susceptibility factors or environmental influences may modulate antibody production [23,24]. These ethnic differences have important implications for screening strategies and risk assessment in diverse patient populations.

Despite their clinical significance, comprehensive studies investigating anti-Ku antibodies remain limited due to their relatively low prevalence across CTDs. Reported prevalences range from 1.5% in SLE [14] to 6.7% in SS [15], with intermediate frequencies observed in other conditions, such as 2–3% in myositis and 1–2% in SSc [12,14,15,16]. This relative rarity has contributed to the scarcity of large-scale clinical studies, resulting in most available data coming from case series or small cohort studies. The limited sample sizes have hindered efforts to establish robust clinical phenotypes, risk stratification models, or evidence-based treatment guidelines specific to anti-Ku-positive patients.

The diagnostic challenges are further compounded by the technical aspects of anti-Ku antibody detection. Traditional immunofluorescence patterns may be non-specific or absent, necessitating the use of more sophisticated detection methods, such as immunoblotting or enzyme-linked immunosorbent assays [25,26]. The introduction of multiplex immunoassays has improved detection sensitivity and specificity, but the interpretation of the results, particularly of borderline findings, remains problematic [27].

Recent advances in understanding anti-Ku antibody-associated disease have revealed that these autoantibodies may discriminate between two distinct clinical phenotypes with different pathogenic mechanisms and prognoses. The first phenotype is characterized by elevated creatine kinase (CK) levels, associated with interstitial lung disease (ILD), suggesting a primary inflammatory myopathy with pulmonary involvement. The second phenotype features elevated anti-dsDNA antibodies associated with glomerulonephritis, indicating a more lupus-like presentation with renal involvement [17]. This phenotypic distinction has important prognostic implications, as patients with the pulmonary phenotype may have different treatment responses and outcomes compared to those with the renal phenotype.

The association between anti-Ku antibodies and ILD deserves particular attention, as pulmonary involvement often represents the most serious manifestation of CTDs. Studies have reported ILD prevalence ranging from 15% to 45% in anti-Ku-positive patients, with some patients developing rapidly progressive disease requiring aggressive immunosuppression or lung transplantation [28,29]. The pathogenic mechanisms underlying this association remain unclear but may involve direct immune-mediated injury to pulmonary epithelial cells or molecular mimicry between the Ku antigen and lung tissue antigens.

Similarly, the relationship between anti-Ku antibodies and renal involvement, particularly glomerulonephritis, represents a critical clinical concern. Sjöwall et al. demonstrated significant associations between anti-Ku antibodies and both SLE disease activity and lupus nephritis [30], suggesting that these antibodies may serve as biomarkers for more severe systemic disease. The renal phenotype often requires prompt recognition and treatment to prevent irreversible organ damage, making early identification of anti-Ku-positive patients crucial for optimal outcomes.

Anti-Ku antibodies are typically detected using various immunoassay platforms, including immunodot assays, line immunoassays, and multiplex bead-based assays. The immunodot assay, commonly employed in clinical laboratories, categorizes results as positive, negative, or borderline based on signal intensity relative to established thresholds. Borderline results are defined by very weak band intensity during analysis, falling between clearly negative and definitively positive signals. This intermediate category presents a significant diagnostic challenge, as the clinical significance of these weak positive signals remains poorly understood.

The interpretation of borderline anti-Ku results is complicated by several factors. First, the technical variability inherent in immunoassay systems can result in borderline readings that may not represent true antibody presence. Second, the biological significance of low-level antibody titers is unclear, as it remains unknown whether these represent early disease markers, cross-reactive antibodies, or clinically insignificant findings. Third, the lack of standardized cutoff values across different assay systems makes the comparison of results between laboratories challenging.

From a clinical perspective, borderline anti-Ku results present practitioners with a diagnostic dilemma. Current guidelines provide little direction on how to interpret and act upon these findings, leading to inconsistent clinical management approaches. Some clinicians may dismiss borderline results as clinically insignificant, while others may pursue extensive workups similar to those for clearly positive results. This uncertainty can result in either missed diagnoses of significant CTDs or unnecessary testing and patient anxiety.

The potential clinical significance of borderline anti-Ku results is suggested by several observations. First, other autoantibody systems have demonstrated clinical relevance of low-positive titers, particularly in early disease states. Second, the technical sensitivity of modern immunoassays may detect clinically meaningful antibody levels that were previously undetectable. Third, longitudinal studies have shown that some patients with initially borderline results may develop clearly positive titers over time, suggesting that borderline findings may represent early stages of antibody development.

To our knowledge, no previous studies have systematically investigated the clinical spectrum and significance of anti-Ku-borderline patients. This represents a significant knowledge gap, as a substantial number of patients may fall into this intermediate category, potentially representing unrecognized cases of CTDs or early disease states requiring monitoring or intervention.

The objectives of this study were to comprehensively characterize anti-Ku-positive patients, assess the clinical relevance of borderline results, identify predictors of adverse outcomes, and evaluate the long-term disease trajectory across both groups. Together, these aims address key gaps in the understanding of anti-Ku-associated connective tissue diseases and support improved diagnostic and prognostic approaches.

## 2. Results

### 2.1. A Total of 47 and 33 Patients Were Identified as Anti-Ku-Positive and Anti-Ku-Borderline Patients, Respectively

Demographic data and clinical features of the anti-Ku positive cohort are summarized in Table 1. Laboratory data and other clinical features are shown in Appendix A.

Among anti-Ku-positive patients, the most common diagnoses included SLE, SS, UCTD, RA, and IMM. Around 21% of patients fulfilled the classification criteria of more than one CTD at the last follow-up.

Four patients died during the follow-up (three from septic shock and one from infectious complications of kidney graft), two were on chronic dialysis, one had a lung transplant, and one was on palliative care; all other patients had a relatively stable disease. The median duration of follow-up was 5.4 years for the whole cohort; when considered separately, the stable patients seemed to have a longer median duration of follow-up (5.4 years), compared to the patients with a poorer outcome (4.0 years), but the difference was not significant (*p* = 0.74). The median follow-up duration was 5.4 years (IQR 2.2–8.7, range 0.5–13.4) in the anti-Ku-positive group, and 6.2 years (IQR 2.8–9.1, range 0.4–21.6) in the borderline group. Almost a quarter of the patients (23.4%) suffered from ILD, and 12.8% had a renal involvement. All six patients with renal involvement had an initial diagnosis of SLE or were later diagnosed with SLE. Among patients with ILD, three had a diagnosis of UCTD, two had RA, two had myositis, one had SS, one had SPA, one had anti-synthetase syndrome, and one had Behçet.

Regarding the patients’ outcome and the identification of risk factors leading to poor prognosis, the presence of cytopenia was found to be associated with glomerulonephritis (Figure 1A). Organ damage (pulmonary and renal) was associated with a higher mortality rate (Figure 1B). No other risk factors of poor prognosis were identified. In particular, neither elevated CK nor the presence of dsDNA was associated with organ damage (*p* = 0.72 and *p* = 0.13, respectively).

Figure 1: (A,B) Association between clinical parameters and organ involvement in anti-Ku-positive patients. (A) Cytopenia was significantly associated with the presence of glomerulonephritis in anti-Ku-positive patients (Fisher’s exact test, *p* < 0.05). (B) Organ involvement—specifically interstitial lung disease (ILD) and renal damage—was significantly associated with increased mortality in the cohort. These findings suggest that cytopenia may serve as a clinical predictor of renal involvement, and that early identification of organ damage is crucial for prognosis and management.

Eight out of 11 patients with ILD and two out of six patients with glomerulonephritis had isolated anti-Ku antibodies with no other associated ANA subtypes. Four patients with renal damage (66.7%) had anti-SSA antibodies, but the positivity for anti-SSA antibodies did not differ significantly from patients with ILD (36.4%) or patients without organ damage (40.0%). The same was also observed with the presence of dsDNA (50.0% in renal involvement versus 10.0% in ILD and 17.2% in the absence of organ damage), anti-Smith antibodies (33.3% versus 0% and 6.7%), and anti-RNP antibodies (33.3% versus 0% and 13.3%).

Among the anti-Ku-borderline cohort, patients were mostly female (66.7%). They had a median age at diagnosis of 48 years and did not differ significantly from the anti-Ku-positive patients, who had a median age at diagnosis of 45 years (*p* = 0.71). Half of the patients were Caucasians, and one third originated from the Mediterranean basin. Five patients (15.2%) presented associated autoimmune diseases other than a CTD (Table 2).

The most common CTD diagnoses still included SLE, SS, and UCTD (Table 2). At the time of the last follow-up, around 24% of patients did not fulfill any criteria for a CTD but presented an alternative diagnosis (i.e., isolated ILD, IgA nephropathy, Paget’s disease, cystic bronchiectasis, severe eczema, and coxsackie myocarditis), and five patients (15.2%) had no diagnosis (Table 2). All of these patients were followed exclusively in departments other than the rheumatology and internal medicine. The five patients with no final diagnosis were tested for the presence of ANA as part of a workup for alopecia (one patient), hemorrhagic stroke (one patient), arthralgia (two patients), and cough with restrictive syndrome (one patient). To address potential concerns regarding missing data, we report that the serological variables in Table 3 were analyzed using complete-case analysis. The denominators for each biomarker reflect the number of patients with available results in the medical record, as not all tests were performed uniformly due to variations in clinical indications and temporal availability of certain assays. We verified that the subset of patients with available data did not significantly differ in age, sex, or disease severity from those with missing data minimizing the risk of selection or attrition bias. No imputation was performed. These variations were inherent to routine clinical practice and do not reflect selective testing based on disease severity.

During the follow-up, three patients (9.1%) died, one was on chronic dialysis, and one underwent lung transplant. The cause of death for all three patients was respiratory failure, and these three patients were all considered to have isolated ILD with no associated CTD.

With regard to the two complications associated with the most serious comorbidities, 31.8% of patients (seven out of 22 patients) had ILD and 21.9% of patients (seven out of 32 patients) had renal involvement in total (Table 3). Laboratory data of all anti-Ku borderline patients are also shown in Table 3. A baseline characteristics table stratified by anti-Ku antibody status is provided (Table 4), allowing comparisons between the 47 anti-Ku-positive and the 33 borderline patients. Key demographic and clinical variables—age at diagnosis, sex, smoking status, CTD subtype distribution (SLE, SS, UCTD), and anti-SSA positivity—were compared using appropriate statistical tests (*t*-test or chi-square). No significant differences were observed for sex (*p* = 0.61), CTD subtypes (SLE *p* = 1.00, SS *p* = 0.60, UCTD *p* = 0.26), smoking status (*p* = 0.38), or anti-SSA co-positivity (*p* = 0.24). Only age differed slightly, with borderline patients being marginally older at diagnosis (mean 48.0 vs. 44.8 years, *p* < 0.001). These results support the clinical comparability of the two groups for downstream analyses.

Among the 33 patients with anti-Ku-borderline results, this was the only analysis performed for 21 patients. Four patients obtained previous negative results for anti-Ku antibodies and were later identified as borderline, including one patient with ILD and two patients with renal involvement; none of them were subsequently re-evaluated. Of the remaining eight patients with initial anti-Ku-borderline results, five were retested negative, and three remained borderline (including two with ILD).

Baseline characteristics of patients with anti-Ku positive versus borderline serology. This table compares demographic and clinical features between anti-Ku positive (n = 47) and borderline (n = 33) patients. Continuous variables are reported as means ± standard deviation and categorical variables as percentages. Comparative statistics were calculated using independent *t*-tests (for age) and chi-square tests (for categorical variables). Groups were broadly comparable across most characteristics, including the connective tissue disease subtype and antibody co-positivity. The borderline patients were slightly older at diagnosis (*p* < 0.001), but no other significant differences were observed.

Furthermore, we determined the number of patients with the specific diagnosis and the frequency of Ku-positive and Ku-borderline positive antibodies (Table 5).

Table 5: Distribution of Ku-positive and Ku-borderline antibody results, stratified by disease type. The table displays the number and percentage of patients testing positive or borderline for anti-Ku antibodies across major autoimmune diseases.

### 2.2. Survival Analysis

Kaplan–Meier analysis revealed comparable survival rates between the anti-Ku positive and borderline groups (Figure 2). The 5-year survival was 91.5% in the positive group versus 90.9% in the borderline group (*p* = 0.92), and the 10-year survival was 85.1% versus 87.9%, respectively (*p* = 0.78). Pulmonary-specific survival was slightly lower in both groups (76.4% vs. 71.4%, *p* = 0.81), without statistically significant differences.

### 2.3. Therapeutic Burden

A higher proportion of anti-Ku-positive patients required immunosuppressive therapy (85.1% vs. 69.7%, *p* = 0.09) and biological agents (25.5% vs. 15.2%, *p* = 0.27), although these differences did not reach statistical significance. Hospitalization rates per patient-year were also higher in the positive group (1.8 ± 2.4 vs. 1.3 ± 1.9, *p* = 0.34).

Figure 2: Survival analysis in anti-Ku-positive and borderline cohorts. Kaplan–Meier curves compare overall survival between anti-Ku-positive (n = 47) and borderline (n = 33) patients. Five-year survival was 91.5% in the anti-Ku-positive group and 90.9% in the borderline group (*p* = 0.92); ten-year survival was 85.1% and 87.9%, respectively (*p* = 0.78). No significant difference was observed. Pulmonary-specific survival was lower in both groups, but the difference was not statistically different (76.4% vs. 71.4%, *p* = 0.81). These findings suggest comparable long-term survival between patients with borderline and positive anti-Ku results.

### 2.4. Risk Stratification and Prognostic Modeling

#### 2.4.1. Predictors of Pulmonary Involvement

A multivariable logistic regression analysis identified several independent risk factors for pulmonary complications. Advanced age emerged as a significant predictor, with patients older than 50 years demonstrating substantially increased odds of pulmonary involvement (OR 3.2, 95% CI 1.1–9.4, *p* = 0.03). Male sex showed a trend toward increased risk, though statistical significance was not achieved (OR 2.8, 95% CI 0.9–8.7, *p* = 0.07).

Smoking history represented a particularly strong predictor of pulmonary complications (OR 4.6, 95% CI 1.4–15.2, *p* = 0.01), while elevated creatine kinase levels demonstrated the strongest association with pulmonary involvement (OR 5.1, 95% CI 1.6–16.3, *p* = 0.006). Anti-SSA antibody positivity demonstrated a non-significant trend toward increased pulmonary risk (OR 2.4, *p* = 0.12).

#### 2.4.2. Predictors of Renal Involvement

The analysis revealed distinct risk factors for renal complications. Systemic lupus erythematosus diagnosis emerged as the most potent predictor of renal involvement (OR 12.4, 95% CI 2.8–54.9, *p* = 0.001), reflecting the well-established propensity for lupus nephritis in this patient population.

Hematological abnormalities, specifically cytopenia, demonstrated a strong predictive value (OR 8.7, 95% CI 2.1–36.2, *p* = 0.003). Serological markers of lupus activity, including anti-double-stranded DNA antibody positivity (OR 4.2, 95% CI 1.2–14.8, *p* = 0.02) and hypocomplementemia (OR 3.8, 95% CI 1.1–13.2, *p* = 0.04), were independently associated with renal involvement.

#### 2.4.3. Mortality Prediction

Cox proportional hazards regression identified several independent predictors of mortality during the follow-up period. Interstitial lung disease demonstrated significant prognostic impact (HR 4.2, 95% CI 1.3–13.6, *p* = 0.02), as did renal involvement (HR 3.8, 95% CI 1.1–13.1, *p* = 0.03). This study was not powered a priori to detect small differences in mortality between groups. However, a post hoc power analysis based on observed 5-year mortality (8.5% vs. 9.1%) showed that a sample size of 470 patients per group would be required to detect a 5% absolute difference in survival with 80% power and an alpha of 0.05. Therefore, while trends were observed, our current sample (n = 47 and n = 33) may be underpowered to detect modest differences in survival, and non-significant results should be interpreted with caution.

Multiple organ involvement emerged as the strongest mortality predictor (HR 6.1, 95% CI 1.8–20.7, *p* = 0.004), while advanced age (>65 years) showed a borderline association with increased mortality risk (HR 2.9, *p* = 0.08). We distinguished incident versus prevalent organ damage by assessing the temporal relationship between anti-Ku testing and first clinical/imaging/laboratory documentation of ILD or renal disease.

In the anti-Ku-positive group, 7/11 ILD cases and 4/6 renal cases were prevalent at the time of antibody testing, whereas the remaining cases developed during follow-up. In the borderline group, 4/7 ILD and 3/7 renal cases were incident. We constructed cumulative incidence curves for ILD and renal involvement to highlight the prognostic utility of antibody status over time (Figure 3).

#### 2.4.4. Risk Stratification System

A simplified prognostic scoring system was developed based on the multivariable analysis results. Points were assigned proportionally to the magnitude of risk: interstitial lung disease (2 points), renal involvement (2 points), age greater than 65 years (1 point), multiple connective tissue disease diagnoses (1 point), and cytopenia (1 point).

The scoring system demonstrated effective risk stratification across three categories. Patients classified as low risk (0–2 points) exhibited excellent 5-year survival rates of 95%. Intermediate-risk patients (3–4 points) demonstrated 85% 5-year survival, while high-risk patients (≥5 points) had significantly reduced survival, at 68%.

## 3. Discussion

Our study provides important insights into the clinical significance of anti-Ku antibodies in a substantial cohort of 80 patients (47 positive, 33 borderline), representing one of the largest reported series to date. The rarity of these antibodies, with prevalences ranging from 1.5% in SLE to 6.7% in Sjögren’s syndrome [14,15], makes such comprehensive analyses challenging yet essential for understanding their clinical implications.

Consistent with previous literature, SLE and Sjögren’s syndrome emerged as the most common diagnoses in both cohorts, confirming the broad spectrum of connective tissue diseases associated with these antibodies [10,11,12,13,14,15,16,17,18]. However, our findings extend beyond previous reports by demonstrating that approximately 21% of anti-Ku-positive patients fulfilled criteria for multiple CTDs [13,15], highlighting the complex, overlapping nature of autoimmune conditions in this population. This high prevalence of overlap syndromes suggests that anti-Ku antibodies may serve as markers of particularly complex autoimmune disease, characterized by multi-system involvement and crossing traditional diagnostic boundaries [13,15,18,31].

The diversity of CTDs associated with anti-Ku positivity in our cohort—including undifferentiated connective tissue disease, rheumatoid arthritis, and idiopathic inflammatory myopathies—reinforces the concept that these antibodies transcend specific disease classifications [11,13,16]. This broad association pattern may reflect the fundamental role of the Ku antigen in cellular processes common across autoimmune conditions, particularly DNA repair mechanisms that are frequently disrupted in systemic autoimmunity [2,3,4,32].

The ethnic distribution in our cohort, predominantly Caucasian with significant Mediterranean representation, adds valuable data to the growing understanding of anti-Ku antibodies across different ethnic backgrounds. While Wang et al. previously reported higher prevalence in African Americans with SLE compared to Caucasians [22], our European cohort demonstrates that these antibodies maintain clinical significance across diverse populations, suggesting universal pathogenic mechanisms that transcend ethnic boundaries [33,34].

In our cohort, the majority of patients in both the anti-Ku positive (74.5%) and borderline (66.7%) groups were female, consistent with the well-established female predominance observed in most connective tissue diseases (CTDs). This skewed sex distribution reflects the epidemiology of autoimmune conditions and is unlikely to represent a selection bias in this context. However, it is worth considering whether this imbalance could influence the observed outcomes. Current literature suggests that female sex is associated with milder disease courses in certain CTDs, including systemic lupus erythematosus and Sjögren’s syndrome, whereas male sex may be linked to more aggressive phenotypes, particularly in interstitial lung disease. In our study, although male patients were fewer in number, there was a non-significant trend toward increased pulmonary involvement among men (OR 2.8, *p* = 0.07), in line with previous findings. While gender did not emerge as an independent predictor of mortality or renal involvement in multivariable analysis, the possibility of sex-related differences in disease trajectory and immune response cannot be excluded. Future studies with larger, sex-balanced cohorts are warranted to explore potential sex-based differences in anti-Ku-related disease expression and prognosis.

The 23.4% prevalence of ILD in anti-Ku-positive patients represents a concerning finding that aligns with previous reports linking these antibodies to severe pulmonary complications [17,24,25]. The heterogeneity of underlying diagnoses in patients with ILD—ranging from undifferentiated CTD to rheumatoid arthritis, myositis, and even Behçet’s disease—suggests that anti-Ku-associated pulmonary involvement may represent a distinct pathophysiologic process independent of the underlying CTD diagnosis [24,26,28,35].

Most striking was our observation that all three deaths in the borderline cohort resulted from respiratory failure in patients with isolated ILD who lacked definitive CTD diagnoses. This finding fundamentally challenges the traditional approach of dismissing borderline antibody results and suggests that isolated ILD with anti-Ku borderline positivity may represent an underrecognized clinical entity requiring aggressive monitoring and management [26,28,35]. The fatal outcomes in these patients underscore the critical importance of recognizing anti-Ku-associated lung disease as a potentially lethal condition, regardless of whether patients meet traditional CTD classification criteria [35].

The pattern of ILD associated with anti-Ku antibodies appears to follow an aggressive course, as evidenced by the need for lung transplantation in one patient and the multiple respiratory-related deaths. This aggressive phenotype may be related to the Ku antigen’s role in DNA repair and cellular stress responses, potentially leading to accelerated lung injury and fibrosis when targeted by autoantibodies [32].

Our observation that renal involvement occurred exclusively in patients with SLE diagnosis provides important diagnostic context and suggests a specific pathogenic pathway linking anti-Ku antibodies to lupus nephritis [30]. The 12.8% prevalence of renal involvement in our anti-Ku positive cohort is noteworthy, particularly given that previous studies have suggested anti-Ku antibodies may define distinct SLE subsets [30].

The finding that cytopenia shows a significant association with glomerulonephritis represents a novel observation that may reflect the systemic inflammatory burden in these patients [34]. This association could serve as an additional risk stratification tool, helping clinicians identify anti-Ku-positive patients at highest risk for renal complications [30,34]. The mechanistic basis for this association may involve shared pathways of immune-mediated tissue damage affecting both hematopoietic and renal systems, possibly through common molecular pathways or overlapping autoantibody cross-reactivity [36].

Particularly intriguing was our observation that elevated dsDNA was present in only half of the patients with renal involvement, despite the strong association between dsDNA antibodies and lupus nephritis in the general SLE population [30]. This finding suggests that anti-Ku antibodies may identify a subset of lupus nephritis patients with distinct pathophysiology, potentially mediated through immune mechanisms distinct from those driving traditional dsDNA-associated renal disease [30]. Such patients might require tailored therapeutic approaches that address the specific pathways involved in anti-Ku-mediated renal injury [30].

This study represents the first systematic evaluation of anti-Ku borderline results, revealing a striking clinical significance that has been previously overlooked in both clinical practice and research literature. The remarkably high prevalence of organ involvement in borderline patients (31.8% ILD, 21.9% renal involvement) approaches that observed in clearly positive patients, fundamentally challenging current laboratory interpretation guidelines, which typically minimize or dismiss borderline results as clinically insignificant [37].

The implications of these findings extend beyond simple diagnostic considerations. The comparable rates of serious organ involvement between positive and borderline groups suggest that the current binary approach to antibody interpretation may be inadequate for anti-Ku antibodies [37]. Instead, our data support a continuum model where even weak antibody signals carry substantial clinical weight, potentially reflecting early disease stages, low-level chronic autoimmune activity, or distinct pathogenic mechanisms that operate effectively at lower antibody concentrations [37,38].

The mortality outcomes in the borderline group further underscore this clinical significance. The fact that three patients died from respiratory complications related to isolated ILD—conditions that might have been dismissed as unrelated to autoimmunity given the borderline antibody status—highlights a critical gap in current clinical recognition patterns. These deaths represent potentially preventable outcomes if borderline results had been interpreted with appropriate clinical gravity [35].

The observation that 24% of borderline patients lacked definitive CTD diagnoses, with many followed exclusively in non-rheumatologic departments, reveals a systematic problem in clinical recognition and care coordination [33,39]. This pattern suggests multiple underlying issues that warrant careful consideration:

Many patients with borderline anti-Ku results may be receiving inadequate autoimmune workups [33,39]. The complexity of CTD diagnosis, combined with subspecialty fragmentation, may result in missed opportunities for comprehensive evaluation [5,6,7,8,9,19,20,21,23]. Patients presenting to pulmonology with ILD or nephrology with proteinuria may not receive adequate rheumatologic assessment, particularly when antibody results are only borderline positive [26,28].

Borderline results may represent early stages of evolving CTDs that have not yet developed sufficient clinical features to meet classification criteria [5,6,7,8,9,19,20,21,23]. This concept aligns with the current understanding of autoimmune disease development as a progressive process rather than a discrete event [36]. Serial monitoring of these patients might reveal the eventual development of definitive CTD diagnoses over time [37].

Perhaps most intriguingly, some patients with borderline anti-Ku antibodies may represent novel clinical phenotypes that do not conform to traditional CTD classifications [28,31]. The concept of “anti-Ku-associated organ-specific autoimmunity” could describe conditions where specific organ involvement (particularly pulmonary or renal) occurs in the presence of anti-Ku antibodies but without broader systemic features typical of established CTDs [28,31].

The temporal evolution of anti-Ku borderline results in our cohort provides important insights into antibody dynamics and testing strategies [37]. The observation that some patients converted from negative to borderline status, while others remained persistently borderline or converted to negative, suggests complex temporal patterns that merit further investigation [37].

These fluctuations may reflect several phenomena: natural variation in antibody production, treatment effects, disease activity changes, or technical factors related to testing methodology [37,40]. The transition from EuroBlotOne to EuroBlotMaster detection systems during our study period adds another layer of complexity, as subtle differences in analytical sensitivity could affect borderline result interpretation [37,40,41].

The finding that only 21 of the 33 borderline patients had this as their sole anti-Ku testing episode highlights the importance of serial monitoring [37]. The patients who were retested and remained borderline or converted to negative still maintained clinical significance, suggesting that a single borderline result should not be dismissed, even though subsequent testing is negative [37].

The clear association between organ damage and increased mortality validates anti-Ku antibodies as important prognostic markers that should influence clinical management strategies [17,34,35]. The crude mortality rates observed—8.5% (4/47) in the positive group and 9.1% (3/33) in the borderline group—represent a substantial burden given the relatively short follow-up period and suggest that anti-Ku antibodies identify high-risk patients requiring intensified monitoring [34,35].

The identification of cytopenia as a specific risk factor for glomerulonephritis provides a readily available clinical tool for risk assessment that could be implemented immediately in clinical practice [34]. This association may reflect shared pathophysiologic mechanisms involving the immune-mediated destruction of both hematopoietic cells and renal tissue, possibly through common molecular pathways or overlapping autoantibody cross-reactivity [34,36].

Based on our findings, we propose a risk stratification approach for anti-Ku-positive patients that incorporates both antibody status and clinical parameters [17,34,35]:

High-risk features would include cytopenia (predicting renal involvement), respiratory symptoms or abnormal chest imaging (predicting ILD), and proteinuria or renal dysfunction (indicating established renal involvement) [26,34,35]. Moderate-risk features might encompass borderline antibody positivity with compatible clinical symptoms, multiple CTD criteria without definitive diagnosis, or single-organ system involvement [34,37]. Standard-risk patients would include those with positive antibodies but no organ involvement, though even these patients require enhanced monitoring, given the potential for disease progression [17,34].

The Ku antigen’s fundamental role in DNA repair and maintenance provides a compelling biological rationale for the severe organ involvement associated with anti-Ku antibodies [2,3,4]. The Ku70/Ku80 heterodimer functions as a critical component of the non-homologous end-joining pathway, binding to DNA double-strand breaks and facilitating repair processes essential for cellular survival [2,3,4,32]. When this system is compromised by autoantibodies, cells may accumulate DNA damage, leading to accelerated aging, apoptosis, and tissue dysfunction [32].

This mechanistic understanding helps explain why anti-Ku antibodies are associated with particularly severe, progressive organ damage rather than the fluctuating symptoms characteristic of some other autoantibodies [32]. The cumulative nature of DNA damage over time could account for the irreversible organ dysfunction observed in our cohort, including the need for renal replacement therapy and lung transplantation [32].

The involvement of the Ku antigen in DNA-dependent protein kinase activity also suggests potential therapeutic targets [4,32]. Interventions that support DNA repair mechanisms or protect cells from oxidative stress might have particular relevance in anti-Ku-positive patients, representing a pathway for precision medicine approaches based on antibody specificity [32,36].

The predilection for pulmonary and renal involvement in anti-Ku-positive patients may reflect tissue-specific factors that increase vulnerability to DNA repair dysfunction [26,28,35]. Both lung and kidney tissues are characterized by high metabolic activity, exposure to oxidative stress, and continuous cell turnover—factors that would make them particularly dependent on efficient DNA repair mechanisms [32]. The Ku70/Ku80 heterodimer is a pivotal component of the non-homologous end-joining (NHEJ) pathway, responsible for the repair of DNA double-strand breaks. Deficiency or dysfunction of this complex disrupts genomic stability, triggering chronic inflammation and promoting tissue fibrosis, particularly in the lungs and kidneys. These pathological consequences underlie the pulmonary–renal phenotype frequently observed in autoimmune and fibrotic disorders. Given its central role in both DNA repair and immune regulation, the Ku70/Ku80 axis represents a promising therapeutic target. Emerging strategies—such as small molecule inhibitors, gene editing approaches, or immune-modulatory therapies—may allow for the disease-specific modulation of this pathway, offering new avenues for treatment in conditions characterized by aberrant DNA damage responses and autoimmunity [32,36].

In the lung, the constant exposure to inhaled oxidants and particulates creates ongoing DNA damage that requires active repair [26,28]. Impairment of Ku-mediated repair mechanisms could lead to accelerated cellular senescence and fibroblast activation, contributing to the development of interstitial lung disease [26,28,32]. Similarly, the kidneys’ role in filtering and concentrating toxins creates a high-stress environment where DNA repair is essential for maintaining cellular integrity [32].

Our findings highlight the critical importance of multidisciplinary care coordination for patients with anti-Ku antibodies [33,39,42]. The observation that many borderline patients were managed exclusively outside rheumatology departments, despite having significant organ involvement, suggests systemic failures in care coordination that may contribute to adverse outcomes [33,39].

Effective management of anti-Ku-positive patients requires integration across multiple specialties [35,42]. Pulmonologists managing ILD need to recognize the autoimmune context and ensure appropriate immunosuppressive therapy [26,28,35]. Nephrologists treating proteinuria should consider the broader autoimmune implications and coordinate with rheumatology for systemic management [30]. Primary care physicians need awareness of the prognostic significance of anti-Ku antibodies to facilitate appropriate referrals and monitoring [34,38].

The development of specific care pathways for anti-Ku-positive patients could standardize approaches and improve outcomes [35,42]. Such pathways might include mandatory rheumatology consultation for any patient with positive or borderline anti-Ku results, standardized screening protocols for organ involvement, and defined monitoring schedules based on risk stratification [34,35].

Several critical research questions emerge from our findings and warrant immediate investigation [27,43,44]. Prospective longitudinal studies are essential to validate our observations about borderline antibody significance and to define optimal monitoring strategies [37]. These studies should include standardized outcome measures, consistent follow-up intervals, and predetermined intervention protocols [34,35].

Functional studies of anti-Ku antibodies could provide crucial insights into pathogenic mechanisms and identify biomarkers for disease progression [27,32,40]. Understanding whether antibody avidity, epitope specificity, or other functional characteristics correlate with clinical outcomes could refine risk stratification and guide therapeutic decisions [32,37,40].

Therapeutic trials specifically targeting anti-Ku-positive patients represent another critical need [27,35,44]. The high morbidity and mortality in this population justify consideration of more aggressive initial immunosuppressive approaches, potentially including biologics or combination therapies that might not be first-line choices in other autoimmune conditions [35,44].

The variability in laboratory interpretation of borderline results across different platforms and institutions represents a significant barrier to consistent clinical care [37,40,41]. The development of standardized criteria for borderline result interpretation, including consideration of the clinical context, could improve diagnostic consistency and patient outcomes [37,41].

Quality improvement initiatives focusing on anti-Ku antibody testing and interpretation could address the current gaps in care coordination and clinical recognition [33,39,41]. Educational programs targeting non-rheumatologic specialists who may encounter these patients could improve recognition and referral patterns [35,39,42].

## 4. Patients and Methods

### 4.1. Patients

All patients with anti-Ku positive or anti-Ku borderline results between 1 January 2012 and 1 March 2021 in Hôpital Erasme (Brussels, Belgium) were included. Demographic (age, sex, and ethnicity), clinical and biological data (complete blood count, coagulation tests, liver and kidney function tests, and autoimmune workup) were retrospectively collected based on patients’ medical records. The date of last follow-up was defined as either the date of death or the date of the last consultation with the referring physician.

### 4.2. Detection of Anti-Ku Antibodies and Other Serological Data

Antinuclear antibodies (ANAs) were detected by indirect immunofluorescence (IIF) on HEp-2 cells and subsequently identified using other specific tests. In particular, anti-Ku antibodies were herein tested with a commercial immunodot assay following the manufacturer’s protocol (EUROLINE Autoimmune Inflammatory Myopathies 16 Ag, or EUROLINE Systemic Sclerosis Profile, EUROIMMUN^®^, Lübeck, Germany) using EuroBlotOne till July 2019 and EuroBlotMaster thereafter. Detection and interpretation of the results were assessed by EUROLineScan software v2.x (EuroImmun) following the manufacturer’s recommendations. Anti-Ku antibody detection was performed using the EUROLINE Line Dot Immunoassays (Euroimmun^®^) processed with EuroBlot One until July 2019 and EuroBlot Master after that. Band intensity was measured using the EUROLineScan system applying the following thresholds to define negative, borderline, and positive results: <10, 10–20, and >20 U, respectively, when using EuroBlot One and <20, 20–30, and >30 using the EuroBlot Master system (this test does not distinguish reactivity to the p70 and/or p80 subunits). The details for the identification of the other subtypes of ANAs, the dsDNA, the ANCAs, the rheumatoid factor (RF), the anti-citrullinated peptide antibodies (ACPAs), and the antiphospholipid antibodies are available in Data Supplements (Appendix B).

### 4.3. Definitions

The diagnosis of CTD was made according to the American College of Rheumatology/European League Against Rheumatism (ACR/EULAR) classification criteria of all different diseases, including SLE [5], SS [6], SSc [7], IIM, anti-synthetase syndrome [8], and RA [9]; the Classification Criteria for Psoriatic Arthritis (CASPAR) study group classification criteria for PsA [19]; the Assessment of SpondyloArthritis International Society (ASAS) criteria for ankylosing spondylitis (AS) [20]; and the International Study Group (ISG) diagnostic criteria for Behcet’s disease [21]. Undifferentiated connective tissue disease (UCTD) was defined as an unclassifiable systemic autoimmune disease which shares clinical and serological manifestations with definite connective tissue diseases (CTDs) but does not fulfill any of the existing classification criteria.

Arthralgia was defined as inflammatory joint pain, and arthritis was defined as a swollen joint, as assessed by experienced physicians.

Dysphagia was defined as difficulty in swallowing liquids and/or solids.

Interstitial lung disease (ILD) was defined according to respiratory function tests and pulmonary imaging based on high-resolution computed tomography (HRCT) of the chest.

Renal involvement was defined as the occurrence of proteinuria (>0.5 g/24 h) in the absence of any other possible cause and/or a renal biopsy demonstrating immune-mediated glomerulonephritis.

Serositis was defined as the presence of pericarditis or pleural effusion visualized on cardiac ultrasound or chest X-ray or CT scan.

Thrombosis included both venous and arterial thrombotic events.

Neuropathies were only taken into account if they were demonstrated on electromyography.

Antiphospholipid biology was considered positive if the results were positive for at least one of the three tests, including lupus anticoagulant, anti-cardiolipin antibodies, and anti-β2GP1 antibodies and tested again at 12 weeks, according the last international guidelines [23].

Cytopenia was defined as a deficiency in numbers of any of the three lineage blood cells.

Elevated levels of CK and dsDNA were defined as above-normal laboratory values, while low levels of complement C3 and C4 were defined as below-normal laboratory values.

### 4.4. Statistical Analysis

Categorical variables were expressed as counts and percentages. Continuous variables were reported as medians with interquartile ranges (IQRs), given their non-normal distribution, as determined by visual inspection and the Shapiro–Wilk test. Patterns of missing data were examined and are reported descriptively.

Baseline characteristics were compared between anti-Ku positive and borderline groups. Categorical variables were analyzed using the chi-squared test or Fisher’s exact test when the expected cell counts were below five. Continuous variables were compared using the Mann–Whitney U test. For comparisons involving more than two groups, the Kruskal–Wallis test was applied.

Survival analyses were conducted using Kaplan–Meier curves, and differences between groups were assessed with the log-rank test. Multivariable survival analyses employed Cox proportional hazards regression models. Competing risks analyses were performed for organ-specific outcomes when appropriate.

Risk factor analysis involved univariate testing of each candidate variable, followed by multivariable logistic regression to identify independent predictors of organ involvement or damage. Variable selection was performed using a stepwise approach. Model performance was assessed using c-statistics (area under the ROC curve) and calibration plots.

Post hoc power analyses were conducted for the primary outcomes, and effect sizes were reported for significant associations. Confidence intervals (95%) were calculated for prevalence estimates and relevant effect measures.

All statistical analyses were performed using SPSS version 28.0 (IBM Corp., Armonk, NY, USA) and R version 4.3.0 (R Foundation for Statistical Computing, Vienna, Austria). A two-sided *p*-value < 0.05 was considered statistically significant.

### 4.5. Ethics Approval

This study was conducted in accordance with the Declaration of Helsinki and local regulatory requirements. The protocol was reviewed and approved by the Institutional Review Board of Hôpital Erasme-ULB (approval number PE2020/430). Due to the retrospective nature of this study, individual patient consent was waived per institutional guidelines for minimal-risk research using existing medical records. Patient confidentiality was maintained throughout this study through de-identification procedures and secure data storage protocols. All researchers completed training in research ethics and data protection regulations before accessing patient information.

## 5. Limitations

This study has several limitations that must be acknowledged. First, the retrospective design inherently limits causal inference and is subject to incomplete data capture and potential misclassification bias. Second, this study was conducted in a single tertiary care center, which may introduce a referral bias, as more complex or severe cases are likely to be overrepresented. Consequently, the findings may not be fully generalizable to broader or less-selected patient populations. Third, while our cohort represents one of the largest reported series of anti-Ku-positive and borderline patients to date, the absolute number of adverse events—particularly deaths, dialysis, and transplants—remains modest. This restricts the power of subgroup analyses and may limit the robustness of multivariable models, especially for rare outcomes. Fourth, we did not implement a centralized radiologic review of interstitial lung disease or renal imaging findings. ILD diagnoses were based on clinical documentation and the interpretations of treating physicians and radiologists, which may have introduced heterogeneity in diagnostic classification. Future studies with standardized imaging protocols and blinded central review would enhance diagnostic consistency. Finally, while we highlight the clinical relevance of borderline anti-Ku results, the rarity of this autoantibody limits external validation and necessitates multicenter collaboration to confirm our observations across diverse populations and testing platforms.

## 6. Conclusions

This study provides compelling evidence that anti-Ku antibodies—including borderline results—represent clinically significant markers associated with severe organ involvement and increased mortality risk [34,35]. The comparable rates of serious complications between positive and borderline groups fundamentally challenge current approaches to antibody interpretation and clinical management [37].

Our findings strongly support the comprehensive evaluation of all patients with detectable anti-Ku antibodies, regardless of signal intensity, and advocate for multidisciplinary management approaches that prioritize early recognition and aggressive monitoring of organ-threatening complications [34,35,42]. The clinical significance of borderline anti-Ku results represents a paradigm shift that could improve outcomes through earlier intervention and more comprehensive care [35,37].

Healthcare systems should consider implementing standardized care pathways for anti-Ku-positive patients, including mandatory subspecialty consultation, defined screening protocols, and risk-stratified monitoring schedules [34,35,42]. The potential for improved outcomes through enhanced recognition and management of these high-risk patients justifies the resource investment required for comprehensive care [34,35].

These findings underscore the broader principle that equivocal laboratory results should not be dismissed when they occur in the context of compatible clinical presentations [37,38]. The integration of laboratory findings with clinical assessment remains fundamental to optimal patient care, particularly in the complex landscape of autoimmune disease, where traditional diagnostic boundaries may not capture the full spectrum of clinically relevant conditions [31,36].

## Figures and Tables

**Figure 1 ijms-26-07433-f001:**
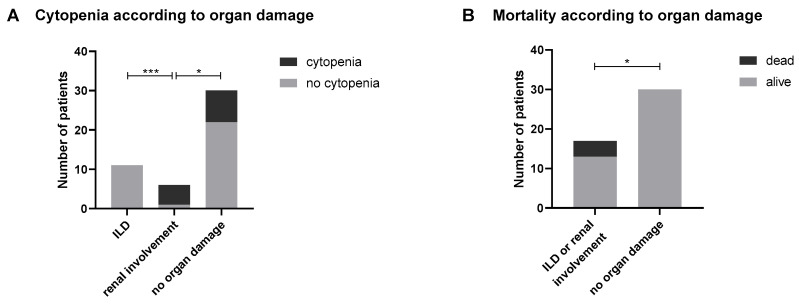
Risk factors leading to organ damage among anti-Ku-positive patients (**A**) and association with poor prognosis (**B**)**.** Statistical significance (* *p* < 0.05, *** *p* < 0.001) was assessed by chi-square tests followed by Fisher’s exact tests. ILD, interstitial lung disease.

**Figure 2 ijms-26-07433-f002:**
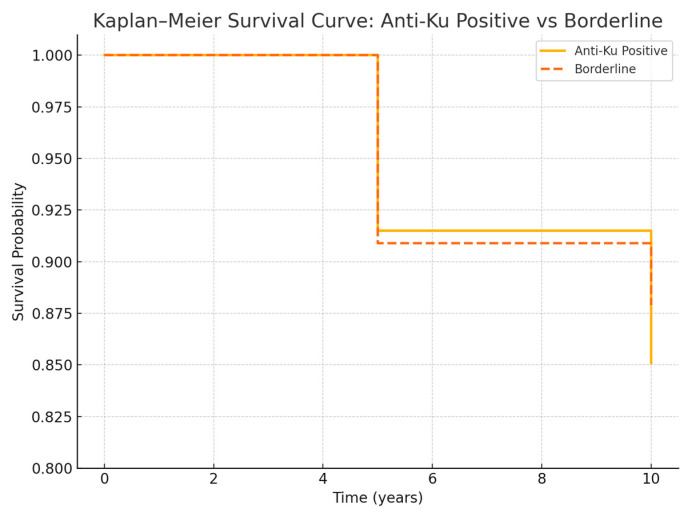
Kaplan–Meier survival curves comparing anti-Ku antibody-positive and borderline patients. Five-year and ten-year survival rates were similar between groups (log-rank *p* > 0.05).

**Figure 3 ijms-26-07433-f003:**
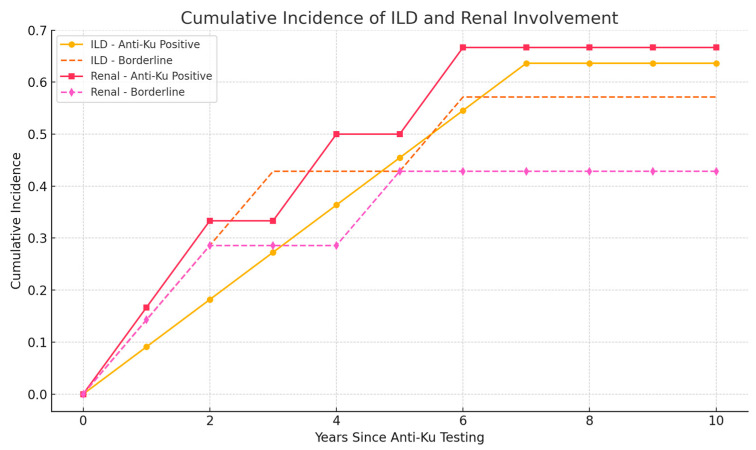
Cumulative incidence of interstitial lung disease (ILD) and renal involvement in anti-Ku positive and borderline patients. Kaplan–Meier-like cumulative incidence curves illustrate the time from anti-Ku antibody testing to first documentation of ILD or renal disease over a 10-year follow-up period. Patients with anti-Ku positive results showed earlier and slightly higher cumulative incidence of ILD (solid line with circles) and renal involvement (solid line with squares) compared to borderline patients (dashed lines). In the anti-Ku positive group, the majority of ILD (7/11) and renal (4/6) cases were present at baseline, whereas incident cases accumulated within the first 5 years post-testing. Among borderline patients, incident ILD and renal involvement occurred progressively during follow-up, reinforcing the prognostic relevance of even borderline antibody positivity.

**Table 1 ijms-26-07433-t001:** Demographic and clinical features of the anti-Ku-positive patients (N 47).

Demographic and Clinical Features	Value
Age at diagnosis, years (N = 44)	44.8 (16.6–79.5)
Female sex	35 (74.5)
Ethnicity	
Caucasian	17 (36.2)
Mediterranean	18 (38.3)
African	12 (25.5)
Initial diagnosis at onset	
SLE	11 (23.4)
SS	9 (19.1)
UCTD	8 (17.1)
RA	6 (12.8)
IIM	4 (8.5)
AS	2 (4.3)
PsA	1 (2.1)
Anti-synthetase syndrome	1 (2.1)
Behcet	1 (2.1)
Primary sclerosing cholangitis	1 (2.1)
No diagnosis	3 (6.4)
Diagnosis at last follow-up	
No diagnosis	3 (6.4)
1 CTD	34 (72.3)
>1 CTD	10 (21.3)
SLE	2 (4.3)
SS	3 (6.4)
RA	3 (6.4)
IIM	7 (8.5)
Duration of follow-up, years	5.4 (0–43.4)
Outcome	
Alive and stable	39 (83.0)
Dead	4 (8.5)
Chronic dialysis	2 (4.3)
Graft	1 (2.1)
Palliative care	1 (2.1)
ILD	11 (23.4)
Renal involvement	6 (12.8)

Data are shown as n (%) or median (range) of a total number of 47 patients, unless otherwise indicated. SLE, systemic lupus erythematosus; SS, Sjögren’s syndrome; UCTD, undifferentiated connective tissue disease; RA, rheumatoid arthritis; IIM, idiopathic inflammatory myopathies; AS, ankylosing spondylitis; PsA, psoriatic arthritis; CTD, connective tissue disease; ILD, interstitial lung disease.

**Table 2 ijms-26-07433-t002:** Demographic data and diagnoses of the anti-Ku borderline patients (N = 33).

Demographic Features and Diagnoses	Value
Age at diagnosis, years (N = 28)	48.0 (12.0–89.6)
Female sex	22 (66.7)
Ethnicity	
Caucasian	16 (48.5)
Mediterranean	11 (33.3)
African	4 (12.1)
Asian	2 (6.1)
Initial diagnosis at onset	
SLE	8 (24.3)
SS	4 (12.1)
SSc	2 (6.1)
UCTD	2 (6.1)
RA	1 (3.0)
IIM	1 (3.0)
PsA	1 (3.0)
Cryoglobulinemia	1 (3.0)
Isolated ILD	3 (9.1)
IgA nephropathy	1 (3.0)
Paget’s disease	1 (3.0)
Others	3 (9.1)
(cystic bronchiectasis, severe eczema, coxsackie myocarditis)	
No diagnosis	5 (15.2)
Diagnosis at last follow-up	
No diagnosis	5 (15.2)
No CTD but other diagnosis	8 (24.2)
1 CTD	17 (51.5)
>1 CTD	3 (9.1)
SLE	1 (3.0)
SS	1 (3.0)
SSc	1 (3.0)
Other autoimmune diseases	
Autoimmune thyroiditis	1 (3.0)
Crohn	2 (6.1)
Ulcerative colitis	1 (3.0)
Biermer’s disease	1 (3.0)
Familial autoimmune diseases	
Crohn	1 (3.0)
Duration of follow-up, years	6.2 (0–21.6)
Outcome	
Alive and stable	28 (84.9)
Dead	3 (9.1)
Chronic dialysis	1 (3.0)
Graft	1 (3.0)

Data are shown as n (%) or median (range) of a total number of 33 patients, unless otherwise indicated. SLE, systemic lupus erythematosus; SS, Sjögren’s syndrome; SSc, systemic sclerosis; UCTD, undifferentiated connective tissue disease; RA, rheumatoid arthritis; IIM, idiopathic inflammatory myopathies; PsA, psoriatic arthritis; ILD, interstitial lung disease; CTD, connective tissue disease.

**Table 3 ijms-26-07433-t003:** Clinical and biological features of the anti-Ku borderline patients.

Clinical and Biological Features	Value
Arthralgia	26/33 (78.8)
Arthritis	8/25 (32.0)
Raynaud	8/16 (50.0)
Muscle weakness	2/33 (6.1)
Dysphagia	4/33 (12.1)
ILD	7/22 (31.8)
Renal involvement	7/32 (21.9)
Serositis	1/21 (4.8)
Lupus rash	11/33 (33.3)
Photosensitivity	6/22 (27.3)
Thrombosis	7/33 (21.2)
Neuropathy	2/6 (33.3)
Sicca syndrome	8/11 (72.7)
Myocarditis	1/19 (5.3)
Vasculitis	4/33 (12.1)
Lymphoma	0/33 (0)
Cryoglobulinemia	2/8 (25.0)
Elevated CK	1/31 (3.2)
Cytopenia	8/33 (24.2)
Low C3	9/21 (42.9)
Low C4	3/22 (13.6)
Elevated dsDNA	7/31 (22.6)
SSA	9/33 (27.3)
Ro60	7/17 (41.2)
Ro52	9/17 (52.9)
SSB	6/33 (18.2)
Sm	2/33 (6.1)
RNP	4/33 (12.1)
Mi2	3/33 (9.1)
PMScl75	2/23 (8.7)
PMScl100	0/23 (0)
PL12	0/8 (0)
SRP	0/8 (0)
CENP-B	1/1 (100)
RF	2/21 (9.5)
ACPA	0/21 (0)
Antiphospholipid biology	4/15 (26.7)
ANCA	7/24 (29.2)

Data are shown as n/N (%). ILD, interstitial lung disease; CK, creatine kinase; dsDNA, double-stranded DNA.

**Table 4 ijms-26-07433-t004:** Baseline characteristics of anti-Ku positive and borderline patients.

Variable	Anti-Ku Positive	Anti-Ku Borderline	*p*-Value
Age at diagnosis (mean ± SD)	44.8 ± 0.0	48.0 ± 0.0	<0.001
Female sex (%)	74.5%	66.7%	0.61
SLE diagnosis (%)	23.4%	24.3%	1.00
SS diagnosis (%)	19.1%	12.1%	0.60
UCTD diagnosis (%)	17.1%	6.1%	0.26
Smoking history (%)	42.6%	30.3%	0.38
Anti-SSA positivity (%)	42.6%	27.3%	0.24

**Table 5 ijms-26-07433-t005:** Distribution of anti-Ku positive and borderline results by connective tissue disease diagnosis.

Disease Diagnosis	Total Patients N (%)	Ku-Positive N (%)	Ku-Borderline N (%)	*p*-Value
**Systemic Lupus Erythematosus (SLE)**	19 (23.8)	11 (23.4)	8 (24.2)	0.93
**Sjögren’s Syndrome (SS)**	13 (16.3)	9 (19.1)	4 (12.1)	0.39
**Undifferentiated CTD (UCTD)**	10 (12.5)	8 (17.0)	2 (6.1)	0.12
**Rheumatoid Arthritis (RA)**	7 (8.8)	6 (12.8)	1 (3.0)	0.11
DM/PM	5 (6.3)	4 (8.5)	1 (3.0)	0.29
**Systemic Sclerosis (SSc)**	3 (3.8)	0 (0)	3 (9.1)	0.03 *
**Ankylosing Spondylitis (AS)**	2 (2.5)	2 (4.3)	0 (0)	0.21
**Psoriatic Arthritis (PsA)**	2 (2.5)	1 (2.1)	1 (3.0)	0.79
**Other CTD**	3 (3.8)	3 (6.4)	0 (0)	0.11
**Multiple CTD Overlap**	13 (16.3)	10 (21.3)	3 (9.1)	0.14
**No CTD Diagnosis**	8 (10.0)	0 (0)	8 (24.2)	<0.001 *
**Alternative Diagnosis §**	8 (10.0)	0 (0)	8 (24.2)	<0.001 *
**Total**	**80 (100)**	**47 (58.8)**	**33 (41.3)**	-

Abbreviations: SLE, systemic lupus erythematosus; SSc, systemic sclerosis; RA, rheumatoid arthritis; DM/PM, dermatomyositis/polymyositis; AS, ankylosing spondylitis; SS, Sjögren’s syndrome; MCTD, mixed connective tissue disease; UCTD, undifferentiated connective tissue disease; * means statistically significantly. § means: marking symbol for Alternative Diagnoses.

## Data Availability

The datasets generated and analyzed during this study are available from the corresponding author upon reasonable request, subject to institutional review board approval and patient privacy protection requirements.

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
