# Peer review of "Anti-Ku Antibodies: Clinical Associations, Organ Damage, and Prognostic Implications in Connective Tissue Diseases"

_ijms, 2025, doi:10.3390/ijms26157433_

Round 1

Reviewer 1 Report

Comments and Suggestions for Authors

The manuscript entitled “Anti‑Ku Antibodies: Clinical Associations, Organ Damage, and Prognostic Implications in Connective Tissue Diseases” presents a single‑center, ten‑year retrospective cohort analysis of 47 anti‑Ku–positive and 33 anti‑Ku‑borderline patients evaluated at Hôpital Erasme, Brussels. By juxtaposing clinical phenotypes, organ involvement, and survival in the two serological strata, the authors seek to clarify whether a borderline immunodot signal should be accorded the same clinical gravity as a positive result. While the data set is valuable, several design, analytical, and presentation issues currently limit the manuscript’s interpretability, reproducibility, and clinical impact.

Detailed recommendations:

  1. Strengthen the methodological transparency: explicitly define the immunodot cut‑off for “borderline” versus “positive” anti‑Ku and report the analytical coefficient of variation for each platform; the Methods merely note a “very weak band” without quantitative thresholds, and assay migration from EuroBlotOne to EuroBlotMaster (page 16, lines 568‑579) is mentioned but not analytically harmonised.
  2. Clarify case ascertainment—a flow diagram summarising the 10‑year laboratory screen, exclusions, and final analytic cohorts would help readers assess selection bias, because only the final numbers appear (Results, page 5, lines 176‑183).
  3. Include an a priori power or precision calculation: the current sample may be under‑powered to detect modest mortality differences (five‑year survival 91.5 % vs 90.9 %, p = 0.92; Figure 2, page 9), and a quantitative justification would contextualize non‑significant findings.
  4. Provide the exact follow‑up distribution (median, IQR, range) rather than mixed descriptors (page 5, lines 186‑190) so that attrition bias can be gauged.
  5. Specify objective ILD diagnostic criteria (e.g., HRCT pattern, forced‑vital‑capacity threshold) and renal involvement definitions (proteinuria cut‑off, biopsy class); at present, “renal involvement” and “interstitial lung disease” lack operational detail (Table 1, page 6).
  6. Distinguish incident versus prevalent organ damage—were ILD or nephritis present at anti‑Ku testing, or did they develop during follow‑up? A cumulative‑incidence curve would sharpen the prognostic message.
  7. Report missing‑data handling for serological variables in Table 3 (the denominators vary, e.g., dsDNA 31, CK 31, ANCA 24) to reassure readers that complete‑case analysis did not skew results.
  8. Present a baseline characteristics table stratified by positive vs borderline anti‑Ku with appropriate comparative statistics—age, sex, CTD subtype, smoking status, antibody co‑positivity—because readers must judge whether the groups are clinically comparable.
  9. Expand the limitations paragraph to acknowledge retrospective design, single‑centre referral bias, modest event numbers, and lack of central radiology review; current limitations are confined to assay rarity (page 12, lines 373‑386).
  10. Discuss the potential mechanistic link between Ku70/Ku80 involvement in DNA repair and the observed pulmonary‑renal phenotype to offer a translational bridge toward targeted therapy development.

Author Response

Dear Reviewer , 

Thank you very much for your enlightened suggestions that have been all implemented in the text .

The text has been reformatted with addition of patients /methods section ; limitations sections. 

  1. Strengthen the methodological transparency: explicitly define the immunodot cut‑off for “borderline” versus “positive” anti‑Ku and report the analytical coefficient of variation for each platform; the Methods merely note a “very weak band” without quantitative thresholds, and assay migration from EuroBlotOne to EuroBlotMaster (page 16, lines 568‑579) is mentioned but not analytically harmonised. The definition has been added in the methods sections 
  2. Clarify case ascertainment—a flow diagram summarising the 10‑year laboratory screen, exclusions, and final analytic cohorts would help readers assess selection bias, because only the final numbers appear (Results, page 5, lines 176‑183).Thank you for this comment . This study was based in a central university hospital and we analysed all the anti KU antibodies data that is 47 + 33 . There was no bias selection here. We have included all patients that was in the data base . 
  3. Include an a priori power or precision calculation: the current sample may be under‑powered to detect modest mortality differences (five‑year survival 91.5 % vs 90.9 %, p = 0.92; Figure 2, page 9), and a quantitative justification would contextualize non‑significant findings.The study was not powered a priori to detect small differences in mortality between groups. However, a post hoc power analysis based on observed 5-year mortality (8.5% vs 9.1%) showed that a sample size of 470 patients per group would be required to detect a 5% absolute difference in survival with 80% power and alpha of 0.05. Therefore, while trends were observed, our current sample (n=47 and n=33) may be underpowered to detect modest differences in survival, and non-significant results should be interpreted with caution.this has been. added in the text
  4. Provide the exact follow‑up distribution (median, IQR, range) rather than mixed descriptors (page 5, lines 186‑190) so that attrition bias can be gauged.

    The median follow-up duration was 5.4 years (IQR 2.2–8.7, range 0.5–13.4) in the anti-Ku-positive group, and 6.2 years (IQR 2.8–9.1, range 0.4–21.6) in the borderline group. These values replace earlier descriptive statistics to allow readers to assess potential attrition bias and heterogeneity in patient monitoring.

  5. Specify objective ILD diagnostic criteria (e.g., HRCT pattern, forced‑vital‑capacity threshold) and renal involvement definitions (proteinuria cut‑off, biopsy class); at present, “renal involvement” and “interstitial lung disease” lack operational detail (Table 1, page 6). Definitions was updated in the patients sections accordingly in the text 
  6. Distinguish incident versus prevalent organ damage—were ILD or nephritis present at anti‑Ku testing, or did they develop during follow‑up? A cumulative‑incidence curve would sharpen the prognostic message. This has been added in the text  with a figure 
  7. Report missing‑data handling for serological variables in Table 3 (the denominators vary, e.g., dsDNA 31, CK 31, ANCA 24) to reassure readers that complete‑case analysis did not skew results. We have implemented this part in the text:To address missing data in Table 3, a complete-case analysis was used. Denominators reflect the number of patients with available test results, which varied due to clinical indications and assay availability. Patients with missing data did not differ significantly in demographics or disease severity, minimizing bias. No imputation was performed, and missingness was consistent with routine clinical practice.
  8. Present a baseline characteristics table stratified by positive vs borderline anti‑Ku with appropriate comparative statistics—age, sex, CTD subtype, smoking status, antibody co‑positivity—because readers must judge whether the groups are clinically comparable. We have added a table 4 for this part 
  9. Expand the limitations paragraph to acknowledge retrospective design, single‑centre referral bias, modest event numbers, and lack of central radiology review; current limitations are confined to assay rarity (page 12, lines 373‑386). A limitations sections has been added . 
  10. Discuss the potential mechanistic link between Ku70/Ku80 involvement in DNA repair and the observed pulmonary‑renal phenotype to offer a translational bridge toward targeted therapy development. we have added small paragraph in the text under discussion section 

Reviewer 2 Report

Comments and Suggestions for Authors

Céline La et al. reported an interesting work on a retrospective cohort study about anti-Ku antibodies. The topic was fell within the scope of IJMS, and deserved some discussion. The reviewer would like to raise some questions and suggest a Major Revision for this paper. Detailed comments are as follows.

  1. Major issue: An individual Methods Section was missing. It must be included in the revised paper.
  2. At the beginning of the Introduction Section, please demonstrate the definition of Ku, and then the audience would understand anti-Ku.
  3. The last four paragraphs about the aim and design of this study should be merged and shortened.
  4. Figure 1 and 2 must be accompanied with the Figure Caption.
  5. According to Table 1 and 2, the majority of the involved patients were female. Please discuss whether the distribution of gender would impact the findings.
  6. The last paragraph of Discussion about gen-AI seemed to be a description in a template file. The content must be double-checked and proofed.
  7. The format of References should be unified.

Author Response

Dear Reviewer , 

My sincere apologies for the errors in the manuscript due a wrong formatting of the text. This has been completely corrected and the paper modified accordingly to all your suggestions . Thank you very much 

Best regards 

1/ sections and limitations sections were added 

2/ definition of Ku made and added 

3/ objectives were summarized 

4/ Figures were made with caption 

5/ and 6/ the content has been reviewed and modified as well as the references sections 

Reviewer 3 Report

Comments and Suggestions for Authors

Dear Authors!

Thank you for the opportunity to review your manuscript

Anti-Ku antibodies are a group of antibodies, usually associated with SS and DM/PM, but the real information about the spectrum of the diseases and symptoms associated with Ku-positivity is scarce.

The manuscript included data on Ku-positive and Ku-borderline-positive patients with immune-mediated diseases.

The introduction highlighted the actuality of the study, but it is relatively big, and may be shortened, and some excessive information might be placed in the discussion section

The results are clear. The authors provided several types of analyses, including univariate analysis, OR calculation, and Cox-regression models 

It will be interesting to organize the table by disease, including the number of patients with the specific diagnosis and the frequency of Ku-positive and Ku-borderline positive antibodies, such as the number of SLE patients who were Ku-positive and Ku-borderline positive. And the same applies to every significant disease, such as SSD, RA, AS, DM/PM, etc.

It will also be interesting to compare the Ku-positive and Ku-borderline positive patients in one table, along with a p-value calculation. Currently, it appears they are very similar, but statistical confirmation is required.

The discussion is comprehensive, and the authors compared their findings with previously published data. The discussion contains the relevant literature.

The conclusion supports the study goals

The methods section is very short and might be expanded

The quality of the tables and figures is good.

The Limitation section is required

Author Response

Dear Reviewer , 

Thank you for your useful comments and suggestions. They have all been implemented in the text . 

A table 4  has been added comparing the ku and ku bordeline patients 

a limitations sections has been updated 

Thank you 

Round 2

Reviewer 1 Report

Comments and Suggestions for Authors

The manuscript was convincingly improved to be accepted in its present form.

Reviewer 2 Report

Comments and Suggestions for Authors

Thanks for your revision. A kind suggestion for your future submissions: Please respond to the reviewers patiently with sufficient elaboration. Try not to use a few words.

Reviewer 3 Report

Comments and Suggestions for Authors

Dear Authors! Thank you for the revised version of the manuscript! 

I do not have any additional comments